# Heterogeneity of Associations between Total and Types of Fish Intake and the Incidence of Type 2 Diabetes: Federated Meta-Analysis of 28 Prospective Studies Including 956,122 Participants

**DOI:** 10.3390/nu13041223

**Published:** 2021-04-07

**Authors:** Silvia Pastorino, Tom Bishop, Stephen J. Sharp, Matthew Pearce, Tasnime Akbaraly, Natalia B. Barbieri, Maira Bes-Rastrollo, Joline W. J. Beulens, Zhengming Chen, Huaidong Du, Bruce B. Duncan, Atsushi Goto, Tommi Härkänen, Maryam Hashemian, Daan Kromhout, Ritva Järvinen, Mika Kivimaki, Paul Knekt, Xu Lin, Eiliv Lund, Dianna J. Magliano, Reza Malekzadeh, Miguel Ángel Martínez-González, Gráinne O’Donoghue, Donal O’Gorman, Hossein Poustchi, Charlotta Rylander, Norie Sawada, Jonathan E. Shaw, Maria Schmidt, Sabita S. Soedamah-Muthu, Liang Sun, Wanqing Wen, Alicja Wolk, Xiao-Ou Shu, Wei Zheng, Nicholas J. Wareham, Nita G. Forouhi

**Affiliations:** 1MRC Epidemiology Unit, Institute of Metabolic Science, Cambridge Biomedical Campus Cambridge, University of Cambridge School of Clinical Medicine, Cambridge CB2 0QQ, UK; Tom.Bishop@mrc-epid.cam.ac.uk (T.B.); Stephen.Sharp@mrc-epid.cam.ac.uk (S.J.S.); Matthew.Pearce@mrc-epid.cam.ac.uk (M.P.); Nick.Wareham@mrc-epid.cam.ac.uk (N.J.W.); 2Department of Population Health, Faculty of Epidemiology and Population Health, London School of Hygiene & Tropical Medicine, London WC1E 7HT, UK; 3Inserm U 1198, Montpellier University, F-34000 Montpellier, France; tasnime.akbaraly@inserm.fr; 4Department of Epidemiology and Public Health, University College London, 1-19 Torrington Place, London WC1E 7HB, UK; m.kivimaki@ucl.ac.uk; 5Postgraduate Program in Epidemiology Faculdade de Medicina, Universidade Federal do Rio Grande do Sul (UFRGS), Porto Alegre 90040-060, Brazil; nataliabordinbarbieri@gmail.com (N.B.B.); bbduncan@ufrgs.br (B.B.D.); maria.schmidt@ufrgs.br (M.S.); 6Department of Preventive Medicine and Public Health, University of Navarra, 31008 Pamplona, Spain; mbes@unav.es (M.B.-R.); mamartinez@unav.es (M.Á.M.-G.); 7CIBERobn, Instituto de Salud Carlos III, 28029 Madrid, Spain; 8Navarra’s Health Research Institute (IdiSNA), 31008 Pamplona, Spain; 9Department of Epidemiology & Data Science, Amsterdam Public Health, Amsterdam Cardiovascular Sciences, Amsterdam UMC—Amsterdam VUMC, 1081 HV Amsterdam, The Netherlands; j.beulens@vumc.nl; 10Julius Center for Health Sciences and Primary Care, University Medical Center Utrecht, 3584 CG Utrecht, The Netherlands; 11MRC Population Health Research Unit, Nuffield Department of Population Health, University of Oxford, Oxford OX3 7LF, UK; zhengming.chen@ndph.ox.ac.uk (Z.C.); huaidong.du@ndph.ox.ac.uk (H.D.); 12Clinical Trial Service Unit and Epidemiological Studies Unit, Nuffield Department of Population Health, University of Oxford, Oxford OX3 7LF, UK; 13Epidemiology and Prevention Group, Center for Public Health Sciences, National Cancer Center, Tokyo 104-0045, Japan; atsushigoto@ucla.edu (A.G.); nsawada@ncc.go.jp (N.S.); 14Department of Public Health Solutions, Finnish Institute for Health and Welfare (THL), FI-00271 Helsinki, Finland; tommi.harkanen@thl.fi (T.H.); paul.knekt@thl.fi (P.K.); 15Digestive Disease Research Center, Digestive Disease Research Institute, Tehran University of Medical Sciences, Tehran 1411713135, Iran; hashemian3@gmail.com (M.H.); dr.reza.malekzadeh@gmail.com (R.M.); H.poustchi@gmail.com (H.P.); 16Biology Department, School of Arts and Sciences, Utica College, Utica, NY 13502, USA; 17Department of Epidemiology, University Medical Center Groningen, University of Groningen, 9713 GZ Groningen, The Netherlands; d.kromhout@umcg.nl; 18Institute of Public Health and Nutrition, University of Eastern Finland, FI-70211 Kuopio, Finland; ritva.jarvinen@uef.fi; 19CAS Key Laboratory of Nutrition, Metabolism and Food Safety, Shanghai Institute of Nutrition and Health, Shanghai, Institutes for Biological Sciences, University of Chinese Academy of Sciences, Chinese Academy of Sciences, Shanghai 200031, China; xlin@sibs.ac.cn (X.L.); sunliang@sibs.ac.cn (L.S.); 20Department of Community Medicine, Pb. 5060, UiT The Arctic University of Norway, 9037 Tromsø, Norway; Eiliv.lund@uit.no (E.L.); charlotta.rylander@uit.no (C.R.); 21The Cancer Registry of Norway, 0379 Oslo, Norway; 22Baker Heart and Diabetes Institute, 75 Commercial Road, Melbourne, VIC 3004, Australia; Dianna.Magliano@bakeridi.edu.au (D.J.M.); jonathan.shaw@bakeridi.edu.au (J.E.S.); 23Department of Nutrition, Harvard T.H. Chan School of Public Health, 665 Huntington Avenue, Boston, MA 02115, USA; 24School of Public Health, Physiotherapy & Sports Science, University College Dublin, Belfield, DO4 Dublin, Ireland; grainne.odonoghue@ucd.ie; 25School of Health & Human Performance, National Institute for Cellular Biotechnology, Dublin City University, Whitehall, DO9 Dublin, Ireland; donal.ogorman@dcu.ie; 26Center of Research on Psychological and Somatic Disorders (CORPS), Department of Medical and Clinical Psychology, Tilburg University, P.O. Box 90153, 5000 LE Tilburg, The Netherlands; S.S.Soedamah@tilburguniversity.edu; 27Institute for Food, Nutrition and Health, University of Reading, Reading RG6 6AR, UK; 28Division of Epidemiology, Department of Medicine, Vanderbilt Epidemiology Center, Vanderbilt-Ingram Cancer Center, Vanderbilt University Medical Center, 2525 West End Avenue, Nashville, TN 37203, USA; wanqing.wen@vanderbilt.edu (W.W.); xiao-ou.shu@vumc.org (X.-O.S.); wei.zheng@vanderbilt.edu (W.Z.); 29Department of Surgical Sciences, Orthopaedics, Uppsala University, 75185 Uppsala, Sweden; Alicja.Wolk@ki.se; 30Institute of Environmental Medicine, Karolinska Institutet, 17177 Stockholm, Sweden

**Keywords:** type 2 diabetes, fish, federated meta-analysis, prospective studies

## Abstract

The association between fish consumption and new-onset type 2 diabetes is inconsistent and differs according to geographical location. We examined the association between the total and types of fish consumption and type 2 diabetes using individual participant data from 28 prospective cohort studies from the Americas (6), Europe (15), the Western Pacific (6), and the Eastern Mediterranean (1) comprising 956,122 participants and 48,084 cases of incident type 2 diabetes. Incidence rate ratios (IRRs) for associations of total fish, shellfish, fatty, lean, fried, freshwater, and saltwater fish intake and type 2 diabetes were derived for each study, adjusting for a consistent set of confounders and combined across studies using random-effects meta-analysis. We stratified all analyses by sex due to observed interaction (*p* = 0.002) on the association between fish and type 2 diabetes. In women, for each 100 g/week higher intake the IRRs (95% CIs) of type 2 diabetes were 1.02 (1.01–1.03, *I*^2^ = 61%) for total fish, 1.04 (1.01–1.07, *I*^2^ = 46%) for fatty fish, and 1.02 (1.00–1.04, *I*^2^ = 33%) for lean fish. In men, all associations were null. In women, we observed variation by geographical location: IRRs for total fish were 1.03 (1.02–1.04, *I*^2^ = 0%) in the Americas and null in other regions. In conclusion, we found evidence of a neutral association between total fish intake and type 2 diabetes in men, but there was a modest positive association among women with heterogeneity across studies, which was partly explained by geographical location and types of fish intake. Future research should investigate the role of cooking methods, accompanying foods and environmental pollutants, but meanwhile, existing dietary regional, national, or international guidelines should continue to guide fish consumption within overall healthy dietary patterns.

## 1. Introduction

The prevalence of type 2 diabetes has been increasing globally, and it is predicted to affect an estimated 700 million people by 2045 [1]. The human and monetary cost of diabetes is vast. Healthy dietary changes are an important way to reverse the current diabetes crisis. Fish consumption has been shown to have cardiometabolic benefits among the general population and diabetes patients [2,3]. Benefits such as improved lipid profile and reduced inflammation have been attributed to the high content of long-chain n-3 fatty acids (LCFAs) [3]. However, the evidence on the benefits of fish intake for the prevention of type 2 diabetes is inconclusive. Systematic reviews and meta-analyses [4,5,6,7,8,9,10] have concluded that the association of fish consumption with diabetes risk differs by geographical location. Studies from North America reported an increased risk [11,12] of type 2 diabetes with fish consumption, while studies from Asia have reported both inverse [13,14] and positive [15,16] associations; studies from Europe show either no risk [17,18] or increased risk [19], with an overall null summary estimate. Types of fish consumed, cooking methods and levels of fish contaminants, which might vary by geographical location, are possible explanations for these heterogeneous findings and whether sex differences may exist is unresolved. Methodological issues such as variation in covariate adjustment may also contribute. 

Previous systematic reviews lack distinctions between types of fish (for example, fatty fish, lean fish, and shellfish), which might underpin the observed differences. Only one of the reviews [5] assessed fish types and showed an inverse association for oily fish consumption, while other types had null associations with type 2 diabetes. However, the results were based on only four studies and were driven by one large study [17]. The evidence on the association between types of fish and type 2 diabetes from individual studies remains ambiguous. For example, lean fish intake was positively associated with type 2 diabetes in the Rotterdam Study [19] but inversely associated in a cohort of Norwegian women [20]. Fatty fish intake was found to be weakly inversely associated with type 2 diabetes in the European Prospective Investigation into Cancer (EPIC)-InterAct study [17], but null associations were reported in other studies [19,20,21]. Similarly, shellfish consumption was inversely associated in the Shanghai Women’s Health Study and the Shanghai Men’s Health Study [13] but positively associated with type 2 diabetes in the EPIC-Norfolk study [21] and a Cohort of Swedish Men [22], while null associations were reported in EPIC-InterAct [21].

Considering the inconsistency of the previous evidence and the potential limitations of a literature-based meta-analysis of published summary results, we used a federated meta-analysis of individual data [23] to investigate the association between the total and types of fish intake and type 2 diabetes across 28 prospective cohort studies, 11 of which did not publish on this association earlier, joined a consortium created as part of the InterConnect project [24].

## 2. Materials and Methods

### 2.1. Populations

InterConnect is a European Commission-funded project, which optimises the use of existing data by enabling cross-cohort analyses within consortia without pooling of data at a central location [24]. For the current research question, 43 studies with information on fish intake and incident type 2 diabetes were invited to join the consortium. These studies were identified by searching published articles on PubMed containing information on type 2 diabetes incidence and dietary fish intake and by reviewing the methodology used in each of the identified cohorts. 28 prospective cohort studies were included in the final collaborative group. Reasons for non-participation of studies varied, including being unable or unwilling to set up a server to allow federated analyses; low priority for the research question; or lacked funding or resources. Of the included studies, 20 studies set up a server to allow federated meta-analysis, while two studies performed analyses locally and sent results; for the remaining six studies, the data were obtained by the approval of data-sharing requests. Characteristics of the participating studies are shown in Appendix A. All cohorts obtained ethical review board approval at the host institution and informed consent from participants.

#### Dietary Assessment

Details of the dietary assessment methods used in the collaborating cohorts are shown in Appendix A. Briefly, 24 cohorts used food frequency questionnaires (FFQ), three cohorts used a dietary history interview, and one cohort used a 24-hour recall. The number of fish consumption items available in each cohort is detailed in Appendix A. We harmonised the variable for total fish intake (g/d, all cohorts) by summing all of the available fish items (g/d, all cohorts). The fish items included fatty, lean, salted, smoked, and dried fish, seafood other than fish, as well as other types of fish and fish products that did not fall into the above categories. For some cohorts, we created total fish by summing saltwater and freshwater fish as well as other fish products. The servings of fish were transformed into g/d of intake if this was not already available. If no information on portion size was available, we considered 120 g to be one serving. The following types of fish (g/d) were harmonised: lean fish (fish with low or very low-fat content in flesh meat); fatty fish (fish with fat content in flesh meat of >4%); seafood other than fish (molluscs and crustaceans); fried fish; salted, smoked, and dried fish; freshwater fish (fish predominantly living in a freshwater habitat); saltwater fish (fish predominantly living in a saltwater habitat). After checking for linearity by comparing associations across quantiles, we present associations as rate ratios per 100 g/week higher fish intake.

### 2.2. Ascertainment of Incident Type 2 Diabetes

To minimise heterogeneity due to different diagnostic criteria, we defined two harmonised outcomes: ‘clinically incident type 2 diabetes’ (primary outcome) and ‘incident type 2 diabetes’ (secondary outcome). For the primary outcome, a confirmed clinical case of type 2 diabetes was considered as fulfilling any one or more of the following criteria: (1) ascertained by linkage to a registry or medical record; (2) confirmed anti-diabetic medication usage; (3) self-report of physician diagnosis or anti-diabetic medication, verified by any of the following: (a) at least one additional source from 1 or 2 above, (b) biochemical measurement (glucose or HbA1c), (c) a validation study with high concordance. For the secondary outcome, which was more inclusive, a case of incident type 2 diabetes was confirmed by any of the following criteria: (1) ascertained by linkage to a registry or medical record; (2) confirmed anti-diabetic medication usage; (3) self-report of physician diagnosis or antidiabetic medication; or (4) biochemical measurement (glucose or HbA1c).

### 2.3. Potential Confounding Factors and Other Covariates

The following factors were considered as potential confounders: age, sex, education, smoking, physical activity, alcohol intake, body mass index (BMI), comorbidities at baseline (baseline diagnosis of any of the following: myocardial infarction, stroke, cancer, or hypertension), energy intake and intake of fibre, red and processed meat, fruit, vegetables, and sugary drinks, family history of diabetes, waist circumference, and fish oil supplements. Details of the specific confounding variables used for each cohort are presented in Appendix A.

### 2.4. Statistical Analyses

The analyses were conducted using R (R Core Team, Vienna, Austria) within the DataSHIELD federated meta-analysis library [25], permitting analyses to be undertaken without the necessity for individual participant data to be transferred and stored at a central location with all the advantages of a traditional individual participant data meta-analysis. For the main analyses, we excluded participants with a diagnosis of any diabetes at baseline (prevalent diabetes), those reporting implausible energy intakes (<500 or >3500 kcal/d for women and <800 or >4200 kcal/d for men) [26] and those with missing values for any of the outcomes, exposures, or confounding factors. The incidence rate ratios (IRRs) and 95% confidence intervals (CIs) for type 2 diabetes were derived using piecewise Poisson regression in each study. The piecewise Poisson regression is available in the DataSHIELD library and is very similar to Cox regression. For the 8 countries of the European Prospective Investigation into Cancer (EPIC)-InterAct case-cohort study, we applied a correction that is analogous to Prentice weighting (weights of 1 for all cases and weights of $\frac{\#\;\mathrm{non}-\mathrm{cases}\;\mathrm{in}\;\mathrm{whole}\;\mathrm{cohort}}{\#\;\mathrm{non}-\mathrm{cases}\;\mathrm{in}\;\mathrm{subcohort}\;}$ for non-cases) for case-cohort studies in survival analyses when using the piecewise Poisson method [27]. Model 1 included age, sex, education, smoking, physical activity, alcohol intake, BMI, and co-morbidities at baseline. Model 2 also included the following dietary factors: energy intake and intake of fibre, red and processed meat, fruit, vegetables, sugary drinks. Some potential confounders (family history of diabetes, waist circumference, and fish oil supplement use) were not available for all cohorts and were only included in sensitivity analyses. Models were fitted within each individual study (or EPIC-InterAct country), and random-effects meta-analysis was used to combine effect estimates and to estimate the degree of heterogeneity (*I*^2^ statistic) using STATA/SE 14.2 (StataCorp, College Station, TX, USA). 

We investigated the effect modification by sex, age, and BMI by including the relevant multiplicative interaction parameter in the models and subsequently combining these parameter estimates across studies. If the combined interaction parameter was statistically significant (*p* < 0.05), analyses were stratified (for sex and for BMI: BMI < 25 kg/m^2^ and BMI ≥ 25 kg/m^2^). Since the heterogeneity of the association between fish intake and type 2 diabetes across different geographical areas was reported in previous meta-analyses [4,5,6,7,8,9,10], we further presented the results by the following geographical regions according to WHO classification [28]: the Americas, including North and South America; Europe; the Eastern Mediterranean; and the Western Pacific, including China, Japan and Australia. There were no studies from the African and South-East Asia Regions. We also tested whether region or age were significant predictors of the association using meta-regression.

## 3. Results

Table 1 shows the characteristics of the participants. After exclusions, 956,122 individuals were included in the analyses. Four cohorts comprised of only women (EPIC-InterAct France; Shanghai Women Health Survey; Women Health Initiative; Norwegian Women and Cancer Study), and three cohorts comprised of only men (Puerto Rico Heart Health Program; Shanghai Men Health Survey; Zutphen Elderly). During follow-up ranging from 4 to 25 years, 48,084 clinically incident cases of type 2 diabetes were recorded for the primary endpoint (n = 49,410 when using the secondary outcome of type 2 diabetes incidence). 

The estimated total fish intake ranged between 3.7 g/d in the Golestan study to 86.1 g/d in the Norwegian Women and Cancer study and tended to be higher in far Eastern (China, Japan), Nordic (Sweden, Norway), and Mediterranean countries (Spain, Italy) (Table 2). Among the different types of fish, lean fish was reported as the most frequently consumed fish type in the Americas and Europe. Shellfish was consumed in appreciable quantities in China, Japan, Sweden, Norway and Spain.

There was evidence of interaction between sex and total fish consumption on type 2 diabetes (*p* = 0.002) but not for BMI or age, so all the results are presented separately for men and women. The results for total fish and clinically incident type 2 diabetes (primary outcome) stratified by sex and geographical region are shown in Figure 1 and Figure 2. There was no association between fish intake and type 2 diabetes for men (Figure 1) in either Model 1 or 2 with low heterogeneity after further adjustments for dietary confounders. In women (Figure 2), the most adjusted model showed a positive association between fish intake and type 2 diabetes, but there was evidence of heterogeneity between studies (*I*^2^ = 61%). Results were similar when we used the secondary outcome of incident type 2 diabetes in both men (Appendix A) and women (Appendix A).

We observed some variation by geographical region (Figure 1), although the region was not a significant predictor in a meta-regression model (*p* = 0.09). Fish intake was not associated with type 2 diabetes incidence in men in any of the different regions. In women, a higher total fish intake was associated with higher type 2 diabetes incidence among women in the Americas, and there was a suggestion, albeit non-significant, of a positive association in Europe and Western Pacific regions. Among the types of fish, there was a positive association between fatty fish consumption and type 2 diabetes incidence in women in the Americas (IRR 1.03, 95%CI: 1.001, 1.064; *I*^2^ = 0), but the association was not significant in other geographical areas (results not shown). 

We analysed the types of fish intake in relation to the type 2 diabetes incidence (Table 3 and Table 4). Both lean and fatty fish were associated with type 2 diabetes risk among women. There was no association between fried fish, salted, dried and smoked fish, saltwater fish, freshwater fish, or shellfish and type 2 diabetes. Only studies from China and Japan contributed to freshwater, saltwater, and salted, dried, and smoked fish; therefore, the sample might have been too small to detect an association.

To further explore heterogeneity, we stratified results by follow-up time (<10 years, ≥10 years). There was no difference in results across strata. Among men with <10 years follow-up RR was 1.00 (95%CI: 0.99, 1.01), *p* > 0.05, *I*^2^ = 56% and with ≥10 years RR was 1.01 (95%CI: 0.998, 1.02), *I*^2^ = 0%.; among women with <10 years follow-up RR was 1.02 (95%CI: 1.01, 1.03), *I*^2^ = 0% and with ≥10 years RR was 1.02 (95%CI: 1.004, 1.04), *I*^2^ = 65%. We performed sensitivity analyses to explore how the inclusion of the family history of diabetes, waist circumference and fish oil supplements use as covariates affected results by conducting analyses only in the subset of studies with any of these variables available (Appendix A). The inclusion of these covariates did not alter the results in the subsets of studies with these variables available. Only six studies had information available on fish oil supplement use (seven among women), so these results were difficult to evaluate in comparison to the main results. To explore whether any one study was primarily responsible for the heterogeneity, we excluded one at a time each study that differed substantially from the overall IRR estimate. In men, after excluding EPIC-InterAct Netherlands, the *I*^2^ value using the most adjusted model dropped from 22% to 0%, with an overall null IRR. In women, after exclusion of EPIC-InterAct France, the *I*^2^ value decreased from 61% to 48%. No other study contributed greatly to heterogeneity.

## 4. Discussion

In this federated meta-analysis of individual-level data of 956,122 adults, including 48,084 confirmed cases of type 2 diabetes, there was a modest positive association between the total, fatty and lean fish intake and type 2 diabetes in women, but not in men. When stratified by region, the positive association of total fish and type 2 diabetes remained for women in the Americas, where it was also positive for fatty fish consumption. Our study provides novel results on the total and types of fish consumption and type 2 diabetes risk across world regions, whereas prior evidence was largely limited to total fish intake using published literature-based results rather than individual-level data.

Our findings of a positive association between combined the total fish and shellfish intake and the risk of type 2 diabetes among women in the Americas are in accordance with previous studies conducted in US cohorts of women [11,12]. The risk of type 2 diabetes comparing the highest with the lowest quantile of fish intake was 1.29 (95%CI: 1.05, 1.57) and 1.32 (95%CI: 0.99, 1.74), respectively, in the Nurses’ Health Study (NHS) and NHS II [11]. The same study found a non-significant positive association for men in the Health Professionals Follow-Up Study (RR = 1.16; 95%CI: 0.96, 1.41) [11]. An increased risk was also found in the Women’s Health Study (RR = 1.49; 95%CI: 1.30, 1.70) [12]. To compare with these previous findings, we also analysed the highest (>3 portions a week) versus lowest (0–1 portion per week) fish consumption. The IRR for highest versus lowest intake for women in the Americas in InterConnect was 1.22 (95%CI: 1.02, 1.46), which is comparable to previous studies. Our finding of a positive association between fatty fish intake and type 2 diabetes in women in the Americas is novel, as the prior US-based studies reported only on total fish intake, not types of fish.

In contrast to previous evidence, we did not find an inverse association between fish intake and type 2 diabetes among Asian cohorts. For instance, in a previous analysis by the Japan Public Health Center-based (JPHC), total fish intake was inversely associated with type 2 diabetes among men but not among women [14]. However, we found no association between fish and type 2 diabetes in the same cohort. Similarly, a negative association was reported between total fish and type 2 diabetes in the Shanghai Women Health Survey (SWHS) [13], but we did not replicate similar findings in our analyses. The discrepancy between our and previously published results might arise from the adjustment for different confounders and from different exposure definitions (e.g., risk per each 100 g/week increment in our study and highest versus lowest categories in published studies). For example, in the SWHS the authors found an inverse association of fish with type 2 diabetes but did not find a dose-response relationship, which suggests a weak association. The longer follow-up time in the current analysis since the publication of these studies a decade ago could also contribute to the observed differences. We also harmonised exposures and outcomes, which reduced heterogeneity, at least among Asian and US cohorts, and might have contributed to the difference in results. Our analysis included a further two Asian studies. In the China Kadoorie Biobank (CKB), we found null associations for men (IRR = 1.02; 95%CI: 0.98, 1.04) and women (IRR = 1.00; 95%CI: 0.98, 1.02); however, recent findings from CKB showed a modest positive association but only amongst urban dwellers [15]. We found a null association in the previously unpublished Nutrition and Health of Aging Population of China Study (NHAPC) for both men and women. 

In women in Europe, there was a suggestion that a higher fish intake was associated with higher type 2 diabetes incidence, but this was not statistically significant. This result is consistent with previous meta-analyses reporting either null or positive associations in Western cohorts [7,8]. We were able to include a larger number of European studies, therefore, increasing our power to detect associations compared to previous reviews. We observed moderately high heterogeneity even after adjustment for potential confounders. However, after the removal of the only study in which fish was negatively associated with type 2 diabetes (EPIC-InterAct France), heterogeneity was decreased among European cohorts, and the positive association between fish and type 2 diabetes became significant. The EPIC-InterAct France cohort, comprised of women teachers, might be healthier than the other included studies. In a previous analysis of EPIC-InterAct France, in contrast to existing evidence, processed meat consumption was negatively associated with type 2 diabetes [29], which might suggest confounding by other unmeasured factors, such as healthier dietary patterns or habits.

The reasons for the small sex difference in IRR that we observed, with a modest positive association between fish intake and type 2 diabetes in women, but a null association in men, are not clear but may include a number of factors such as different dietary patterns, fish cooking methods or by types of fish consumed among men and women as well as by residual confounding. We further explored the relationship between types of fish and type 2 diabetes. For several types of fish, the heterogeneity of association with type 2 diabetes across cohorts was lower (*I*^2^ ranging from 0–47%) than for total fish (*I*^2^ 61%), and both lean and fatty fish, the most commonly consumed types of fish, were associated with a higher risk of type 2 diabetes among women. For seafood and fried fish, the heterogeneity across cohorts was higher (*I*^2^ 74% and 64%, respectively), and there was no significant association of these fish types with type 2 diabetes. Another possible explanation for the observed sex difference may be differences in the degree of measurement error. Dietary under-reporting differs according to sex and age, increasing with older age, and it may be more frequent in women than in men [30,31]; therefore, the degree of reporting bias in our analyses might be different for men and women. However, foods perceived as healthy, such as vegetables and fish, are less likely to be underreported [32]. Reverse causation might have also been an issue if post-menopausal women at greater risk of cardiovascular disease were advised to consume more fish. 

Other not measured factors, such as environmental contaminants, might also have contributed to the geographical and sex discrepancy in the risk. Certain fish contaminants, including persistent organic pollutants (POP), methyl mercury, polychlorinated biphenyls (PCBs), and chlorinated pesticides, which vary by geographical location, have been associated with an increased risk of type 2 diabetes [33,34,35,36,37]. Sex differences in the association between certain POPs and type 2 diabetes have been reported in several studies, with positive associations found among women but not men [38,39,40,41,42]. A possible explanation for sex differences in the association of POPs with type 2 diabetes might be the higher body fat composition in women, with consequent higher storage of lipophilic organic pollutants [38]. Although women tend to have lower blood concentrations of lipophilic pollutants, as these substances pass on to the offspring through breastmilk [43], the year in which a woman gave birth as well as parity affect this process [44]. Older cohorts of women, who are also at higher risk of type 2 diabetes, might accumulate more POPs than younger women because they gave birth before POPs’ bans and regulations, such as the Montreal Protocol and the Stockholm Convention, were introduced from the 1970s onwards [45]. Contaminant measurements were not available in our analysis; therefore, we could not test this hypothesis, but future such research is warranted. 

Another potential explanation for the sex difference may include differential status and metabolization of omega-3 fatty acids, vitamin D, and selenium, which are important nutrient components of fish. Sex-specific effects of vitamin D in the pathogenesis of type 2 diabetes have been reported. A few studies [46,47,48] reported an inverse relationship between 25(OH)D serum levels and fasting insulin, insulin production, and cardiometabolic risk only in men. This sex disparity may result from differences in endogenous sex hormones [49]. The active metabolite of vitamin D, 1,25-dihydroxyvitamin D, is involved in steroid hormone production, and higher levels are linked to high testosterone in both men and women [50,51]. High testosterone levels have been associated with a higher risk of type 2 diabetes among women but decreased the risk among men [49,52]. A similar sexually dimorphic relationship was found for selenium and lipid metabolism, possibly due to sex differences in selenium uptake and selenoprotein expression [53]. High serum selenium has been positively correlated with waist circumference, systolic blood pressure, triglycerides, fasting glucose, and homeostatic model assessment insulin resistance (HOMA-IR) in women, but only with fasting glucose and HOMA-IR in men [54]. Sex differences in circulating concentrations of omega-3 fatty acids have also been reported, with a suggestion of women’s higher synthesis of long-chain polyunsaturated fatty acids from shorter chain n-3 fatty acids [55,56,57]. A recent systematic review suggested a higher risk of type 2 diabetes and insulin resistance measures with omega-3 fatty acids supplementation at high doses above 4.4 g/d [58]. Therefore, an increase in risk with omega-3 supplementation or possibly also with very high fish intake might be apparent only in women, who may already have higher circulating omega-3 fatty acid levels than men. However, these explanations are speculative and should be the subject of further investigation in studies adequately designed for such research.

A major strength of our study was the use of individual participant data from 28 cohort studies, which contributed the largest number to date of incident type 2 diabetes cases in an analysis of fish and type 2 diabetes and a large variation in fish intake estimates across different geographical locations. The large sample also enabled us to interrogate potential interaction by sex, unmasking a sex difference in the association between fish intake and type 2 diabetes. We were also able to investigate the association of different types of fish with type 2 diabetes, unlike prior research. Unlike previous literature-based meta-analyses, we harmonised exposures and outcomes and consistently adjusted for the same confounders, which reduced heterogeneity and enhanced comparability across studies. Furthermore, consortia-based individual-level meta-analyses reduce the risk of publication bias; specifically, we included 11 cohorts that had requisite data but had not previously published on this topic. Our federated meta-analysis approach overcame constraints of the physical pooling of data due to governance or ethical and resource issues. 

Our study had some limitations. Although we adjusted for a range of potential socio-demographic, lifestyle, and dietary confounders, we cannot exclude the possibility of residual confounding due to unmeasured or imprecisely measured factors, which is inherent to observational studies. A further limitation of our analysis is the risk of measurement error in estimating fish intake. The majority of the studies used food frequency questionnaires, and only a few used more precise methods such as diet history interviews. Although the comparability across studies was greatly enhanced through harmonisation, heterogeneity due to different assessments of fish intake and type 2 diabetes, as well as other variables, cannot be ruled out. Despite the careful harmonisation of dietary exposure variables, the wide difference in dietary habits across geographical areas could not be fully captured. It is possible that the positive association between fish and diabetes in women in North America might be due to other foods consumed with fish among this population. Another limitation was the lack of available data on whether the fish consumed was farmed or wild. Our analyses assumed a linear association between exposure and outcome because of the inability to order a pooled data set, and hence non-linear effects could not be examined. Some prior large studies that had previously been published on the association between total fish intake and type 2 diabetes incidence were not included in our analysis (for instance, NHS, NHS II, and Health Professionals Follow-Up Study). However, we are reassured that our findings of a positive association between total fish in women in US-based North American studies were consistent with the prior publications from this region. We attempted to bring together data from diverse geographical locations but could not include any cohorts from some world regions such as Africa and South Asia because of a lack of prospective studies, highlighting this important gap in global health research.

## 5. Conclusions

In summary, we found evidence of a neutral association between total fish intake and type 2 diabetes in men, but there was a modest positive association among women with heterogeneity across studies, which was partly explained by geographical location and types of fish intake. Compared to previous meta-analyses that may be subject to publication bias, our approach facilitated the inclusion of cohorts with data that were previously unpublished on this topic and optimised comparability across studies by harmonisation and consistent confounder adjustment. The reasons for the observed modest positive association between fish intake and type 2 diabetes in women in some Western settings are unclear and require further investigation, including an understanding of the accompanying foods and overall dietary patterns within which fish is consumed, as well as cooking methods and environmental pollutants. It is important to highlight that until findings from further research are available, the existing regional, national, or international guidelines on fish consumption should continue to guide fish consumption among individuals and populations. 

## Figures and Tables

**Figure 1 nutrients-13-01223-f001:**
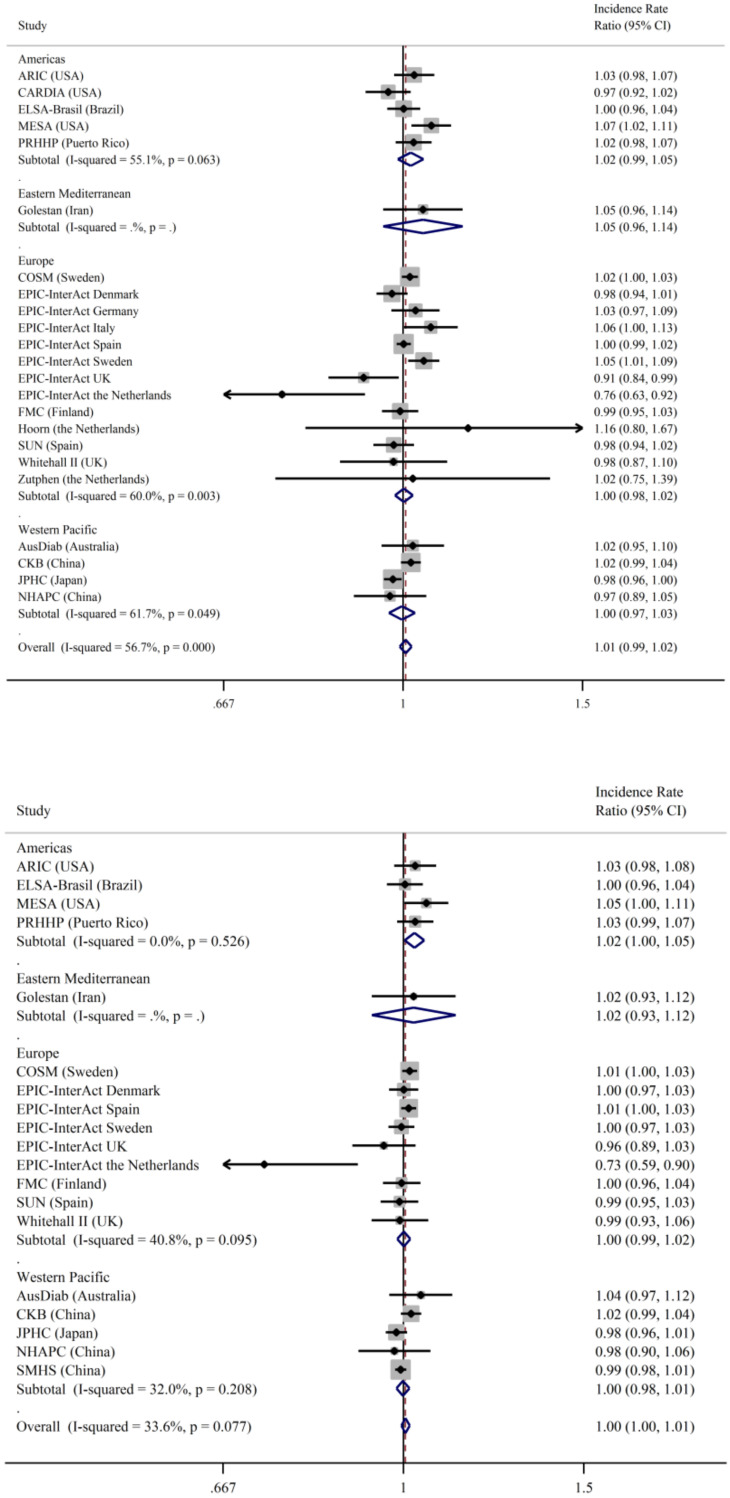
Incidence rate ratios and 95% confidence intervals for the association between the consumption of total fish (per 100 g/day) and incident type 2 diabetes (primary outcome) in men in the InterConnect project. Model 1 (upper panel) adjusted for age, education, smoking, physical activity, alcohol intake, BMI, and comorbidities at baseline. Model 2 (lower panel) was additionally adjusted for dietary factors: energy intake, intake of fibre, red and processed meat, fruit, vegetables, and sugary drinks. ARIC—Atherosclerosis Risk in Communities; ELSA Brasil—Brazilian Longitudinal Study of Adult Health; CARDIA—Coronary Artery Risk Development in Young Adults Study; MESA—Multi-Ethnic Study of Atherosclerosis; PRHHP—Puerto Rico Heart Health Program; FMC—Finnish Mobile Clinic Health Examination Survey; COSM—Cohort of Swedish Men; SUN—Seguimiento Universidad de Navarra (University of Navarra Follow-up); AusDiab—Australian Diabetes, Obesity and Lifestyle Study; CKB—China Kadoorie Biobank; JPHC—Japan Public Health Center-based; NHAPC—Nutrition and Health of Aging Population of China Study; SMHS—Shanghai Men Health Study.

**Figure 2 nutrients-13-01223-f002:**
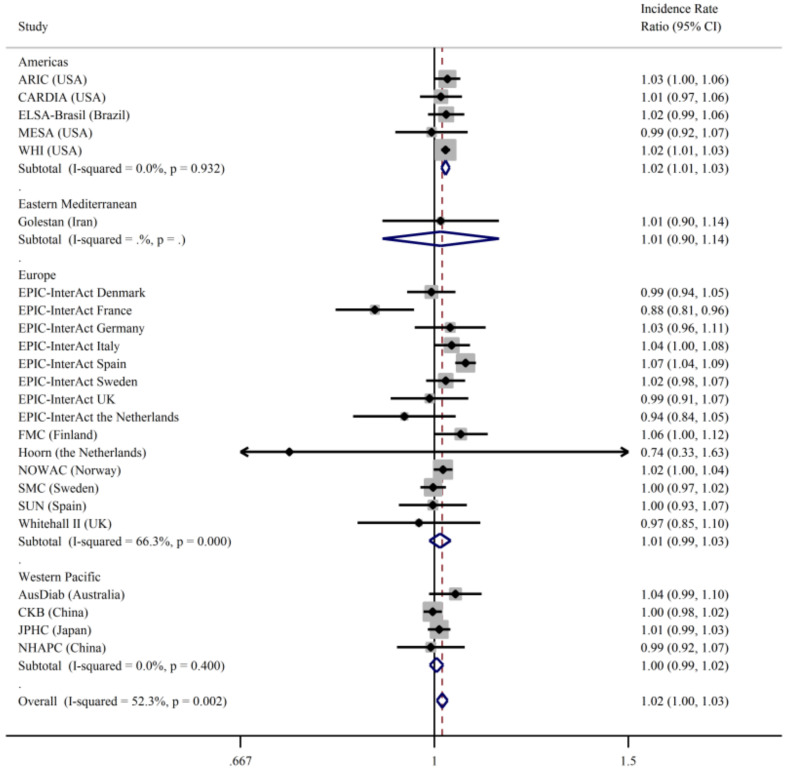
Incidence rate ratios and 95% confidence intervals for the association between the consumption of total fish (per 100 g/day) and incident type 2 diabetes (primary outcome) in women in the InterConnect project. Model 1 (upper panel) adjusted for age, education, smoking, physical activity, alcohol intake, BMI, comorbidities at baseline. Model 2 (lower panel) was additionally adjusted for dietary factors: energy intake, intake of fibre, red and processed meat, fruit, vegetables, sugary drinks. ARIC—Atherosclerosis Risk in Communities; CARDIA—Coronary Artery Risk Development in Young Adults Study; ELSA-Brasil—Brazilian Longitudinal Study of Adult Health; MESA—Multi-Ethnic Study of Atherosclerosis; WHI—Women Health Initiative; FMC—Finnish Mobile Clinic Health Examination Survey; NOWAC—Norwegian Women and Cancer; SMC—Swedish Mammography Cohort; SUN—Seguimiento Universidad de Navarra (University of Navarra Follow-up); AusDiab—Australian Diabetes, Obesity and Lifestyle Study; CKB—China Kadoorie Biobank; JPHC—Japan Public Health Center-based; NHARC—Nutrition and Health of Aging Population of China Study; SWHS—Shanghai Women Health Survey.

**Table 1 nutrients-13-01223-t001:** Participant characteristics in the cohorts participating in the InterConnect project on the association between fish consumption and type 2 diabetes.

Cohort (Country)	Total (N)	Women (%)	New Type 2 Diabetes Cases (n) Primary	New Type 2 Diabetes Cases (n) Secondary	Mean (SD) Age (Years)	Median (IQR) Follow-Up Time (Years)	Mean (SD) BMI (kg/m^2^)
Americas							
ARIC (US)	9654	56	723	2003	53.7 (5.6)	11.8 (8.8, 23.6)	27.1 (4.9)
ELSA Brasil (Brazil)	11,351	57	338	957	51.6 (8.9)	3.8 (3.4, 4.0)	26.7 (4.5)
CARDIA (US)	3920	59	198	198	24.9 (3.5)	25.0 (19.0, 25.0)	24.3 (4.7)
MESA (US)	4669	54	228	674	61.4 (10.1)	4.0 (4.0, 5.0)	27.9 (5.2)
PRHHP (Puerto Rico)	6977	0	310	825	54.1 (6.5)	5.0 (5.0, 5.0)	24.9 (3.8)
WHI (US)	86,296	100	10,233	10,233	63.6 (7.3)	11.8 (7.8, 13.6)	27.1 (5.6)
Eastern Mediterranean							
Golestan (Iran)	9932	52	532	1148	51.2 (7.8)	4.2 (3.6, 5.6)	26.7 (5.2)
Europe							
EPIC-InterAct Denmark	3896	44	1970	1970	56.9 (4.4)	10.3 (6.3, 11.6)	27.3 (4.5)
EPIC-InterAct France	795	100	257	257	56.9 (6.5)	9.2 (7.2, 10.5)	24.5 (4.6)
EPIC-InterAct Germany	3448	51	1505	1505	52.4 (8.3)	9.5 (4.8, 11.2)	27.6 (4.8)
EPIC-InterAct Italy	3112	65	1271	1271	51.4 (7.7)	10.8 (6.8, 12.9)	27.3 (4.8)
EPIC-InterAct the Netherlands	2067	83	741	741	54.1 (10.0)	11.1 (6.4, 12.6)	26.6 (4.5)
EPIC-InterAct Spain	5584	57	2354	2354	50.3 (7.8)	12.4 (8.9, 13.6)	29.3 (4.5)
EPIC-InterAct Sweden	3439	55	1574	1574	58.4 (7.4)	12.0 (9.3, 13.6)	26.8 (4.4)
EPIC-InterAct UK	1858	53	608	608	58.3 (10.5)	10.5 (6.3, 12.2)	26.9 (4.4)
FMC Health Examination (Finland)	9057	49	481	481	39.0 (15.5)	24.2 (22.5, 25.7)	24.7 (4.1)
Hoorn (the Netherlands)	1206	54	16	93	60.0 (6.7)	6.4 (6.1, 6.7)	26.1 (3.1)
NOWAC (Norway)	34,547	100	560	672	49.8 (5.8)	6.0 (6.0, 7.0)	22.4 (3.5)
COSM and SMC (Sweden)	54,571	46	5339	5432	59.9 (9.0)	18.0 (18.0, 18.0)	25.2 (3.4)
SUN (Spain)	19,261	60	142	142	37.6 (12.0)	10.1 (5.9, 12.6)	23.5 (3.5)
Whitehall II (UK)	4554	29	368	632	49.7 (5.9)	16.1 (15.4, 16.5)	25.2 (3.6)
Zutphen Elderly (the Netherlands)	475	0	11	62	70.9 (4.7)	10.1 (5.3, 10.3)	25.6 (2.8)
Western Pacific							
AusDiab (Australia)	6017	56	184	363	49.9 (12.3)	11.7 (5.1, 12.2)	27.5 (4.6)
CKB (China)	482,588	59	9601	9601	51.1 (10.6)	7.2 (6.3, 8.1)	23.5 (3.3)
JPHC (Japan)	50,054	55	801	801	56.1 (7.6)	5.0 (5.0, 5.0)	23.4 (2.9)
NHAPC (China)	932	57	178	225	58.3 (6.0)	6.0 (6.0, 6.0)	24.5 (3.3)
SMHS (China)	61,250	0	2976	2.976	55.3 (9.7)	5.6 (5.0, 6.0)	23.7 (3.0)
SWHS (China)	74,710	100	4585	4585	52.6 (9.0)	10.2 (9.2, 10.8)	24.0 (3.4)

ARIC—Atherosclerosis Risk in Communities; ELSA Brasil—Brazilian Longitudinal Study of Adult Health; CARDIA—Coronary Artery Risk Development in Young Adults Study; MESA—Multi-Ethnic Study of Atherosclerosis; PRHHP—Puerto Rico Heart Health Program; WHI—Women Health Initiative; FMC—Finnish Mobile Clinic Health Examination Survey; NOWAC—Norwegian Women and Cancer; COSM—Cohort of Swedish Men; SMC—Swedish Mammography Cohort; SUN—Seguimiento Universidad de Navarra (University of Navarra Follow-up); AusDiab—Australian Diabetes, Obesity and Lifestyle Study; CKB—China Kadoorie Biobank; JPHC—Japan Public Health Center-based; NHAPC—Nutrition and Health of Aging Population of China Study; SMHS—Shanghai Men Health Study; SWHS—Shanghai Women Health Study.

**Table 2 nutrients-13-01223-t002:** The consumption of total and types of fish in the InterConnect Project by region and cohort.

Cohort (Country)	Total Fish	Fatty Fish	Lean Fish	Seafood	Fried Fish g	Salted, Dried Smoked, Fish	Saltwater Fish	Freshwater Fish
**Americas**								
ARIC (US)	26.9 (18.1, 48.5)	1.9 (1.9, 7.7)	7.7 (1.9, 16.4)	1.8 (1.8, 7.6)	NA	NA	NA	NA
ELSA-Brasil (Brazil)	33.0 (18.0, 58.0)	NA	NA	0.0 (0.0, 3.0)	0.0 (0.0, 12.0)	NA	NA	NA
CARDIA (US)	34.4 (9.2, 80.5)	0.0 (0.0, 0.0)	19.0 (0.0, 46.0)	3.5 (0.0, 23.0)	0.0 (0.0, 0.0)	NA	NA	NA
MESA (US)	24.3 (11.7, 47.2)	3.5 (0.0, 9.2)	3.5 (0.0, 9.2)	1.7 (0.0, 4.6)	3.5 (0.0, 9.2)	NA	NA	NA
PRHHP (Puerto Rico)	0.0 (0.0, 0.0)	NA	NA	0 (0, 0)	NA	NA	NA	NA
WHI (US)	23.0 (11.8, 40.8)	0.0 (0.0, 5.9)	3.9 (0.0, 9.2)	0.0 (0.0, 5.9)	0.0 (0.0, 3.9)	NA	NA	NA
**Eastern Mediterranean**								
Golestan (Iran)	3.7 (0.8, 10.2)	0.0 (0.0, 1.6)	2.2 (0.1, 7.4)	NA	0.0 (0.0, 3.1)	0.0 (0.0, 0.0)	3.0 (0.6, 8.9)	0.0 (0.0, 0.0)
**Europe**								
EPIC-InterAct Denmark	36.6 (26.0, 57.3)	11.7 (6.9, 18.5)	15.4 (9.9, 23.4)	1.7 (0.9, 4.2)	6.1 (3.0, 12.9)	NA	NA	NA
EPIC-InterAct France	30.9 (18.6, 47.2)	8.9 (4.0, 16.2)	10.2 (0.0, 20.1)	0.0 (0.0, 4.6)	0.0 (0.0, 0.0)	NA	NA	NA
EPIC-InterAct Germany	17.2 (9.0, 29.0)	0.0 (0.0, 1.2)	0.0 (0.0, 0.0)	0.0 (0.0, 0.0)	4.2 (1.6, 6.9)	NA	NA	NA
EPIC-InterAct Italy	25.3 (14.2, 40.8)	8.1 (3.7, 14.9)	5.1 (1.2, 12.2)	2.8 (0.9, 6.8)	0.0 (0.0, 0.0)	NA	NA	NA
EPIC-InterAct the Netherlands	8.4 (3.4, 16.1)	1.4 (0.6, 3.6)	1.5 (0.5, 3.3)	0.7 (0.3, 1.7)	3.0 (1.0, 6.6)	NA	NA	NA
EPIC-InterAct Spain	56.5 (35.2, 85.5)	11.3 (2.8, 24.4)	25.7 (10.2, 49.1)	3.6 (0.0, 8.6)	0.0 (0.0, 0.0)	NA	NA	NA
EPIC-InterAct Sweden	36.7 (19.2, 57.6)	2.3 (0.0, 16.3)	0.0 (0.0, 16.6)	1.7 (0.0, 6.1)	2.4 (0.0, 8.3)	NA	NA	NA
EPIC-InterAct UK	31.6 (18.1, 45.7)	8.1 (0.0, 16.1)	17.9 (8.1, 26.2)	0.0 (0.0, 4.2)	0.0 (0.0, 12)	NA	NA	NA
FMC (Finland)	19.0 (9.0, 35.0)	6.3 (2.0, 15.0)	7.0 (2.0, 15.0)	0.0 (0.0, 0.0)	4.5 (0.6, 9.3)	4.0 (0.7, 11.0)	5.5 (1.7, 12.8)	7.0 (2.0, 17.0)
Hoorn (the Netherlands)	12.0 (1.0, 25.0)	1.0 (0.0, 8.0)	3.5 (0.0, 10.0)	0.0 (0.0, 0.0)	0.0 (0.0, 0.0)	NA	NA	NA
NOWAC (Norway)	86.1 (57.2, 123.5)	11.4 (4.8, 21.4)	23.6 (10.9, 40.7)	3.5 (0.0, 3.5)	NA	NA	NA	NA
COSM and SMC (Sweden)	29.0 (20.0, 41.0)	8.0 (6.0, 15.0)	10.0 (8.0, 25.0)	4.0 (3.0, 5.0)	12.3 (4.1, 16.4)	NA	NA	NA
SUN (Spain)	85.7 (56.9, 128.6)	21.4 (10.0, 64.3)	31.4 (21.4, 74.3)	16.7 (10.0, 20.7)	NA	0.0 (0.0, 3.3)	NA	NA
Whitehall II (UK)	35.0 (17.5, 52.5)	8.7 (0.0, 17.5)	17.5 (8.7, 26.2)	0.0 (0.0, 8.7)	0.0 (0.0, 8.7)	NA	NA	NA
Zutphen Elderly (the Netherlands)	13.0 (0.0, 29.0)	0.0 (0.0, 8.0)	8.0 (0.0, 20.0)	0.0 (0.0, 0.0)	0.0 (0.0, 13.0)	0.0 (0.0, 3.0)	13.0 (0.0, 29.0)	0.0 (0.0, 0.0)
**Western Pacific**								
AusDiab (Australia)	25.3 (13.7, 44.0)	NA	NA	NA	3.3 (1.5, 10.3)	NA	NA	NA
CKB (China)	8.2 (1.2, 32.8)	NA	NA	NA	NA	NA	NA	NA
JPHC (Japan)	79.1 (50.0, 121.2)	27.0 (15.3, 48.9)	8.0 (0.0, 20.0)	10.7 (7.0, 18.3)	NA	11.7 (4.4, 25.0)	NA	40.1 (24.0, 65.6)
NHAPC (China)	41.0 (20.7, 69.8)	NA	NA	5.5 (1.9, 15.9)	NA	1.4 (0.5, 3.6)	9.8 (3.3, 21.4)	14.3 (6.6, 28.6)
SMHS (China)	38.4 (21.0, 66.1)	NA	NA	11.5 (2.8, 15.0)	NA	NA	21.5 (6.0, 26.2)	16.5 (3.8, 21.0)
SWHS (China)	8.9 (1.4, 35.7)	NA	NA	10.0 (2.3, 12.5)	NA	NA	20.4 (3.6, 26.2)	17.5 (4.2, 21.0)

Values are median and interquartile range. ARIC—Atherosclerosis Risk in Communities; ELSA-Brasil—Brazilian Longitudinal Study of Adult Health; CARDIA—Coronary Artery Risk Development in Young Adults Study; MESA—Multi-Ethnic Study of Atherosclerosis; PRHHP—Puerto Rico Heart Health Program; WHI—Women Health Initiative; FMC—Finnish Mobile Clinic Health Examination Survey; NOWAC—Norwegian Women and Cancer; COSM—Cohort of Swedish Men; SMC—Swedish Mammography Cohort; SUN—Seguimiento Universidad de Navarra (University of Navarra Follow-up); AusDiab—Australian Diabetes, Obesity and Lifestyle Study; CKB—China Kadoorie Biobank; JPHC—Japan Public Health Center-based; NHAPC—Nutrition and Health of Aging Population of China Study; SMHS—Shanghai Men Health Survey; SWHS—Shanghai Women Health Survey.

**Table 3 nutrients-13-01223-t003:** Adjusted incidence rate ratios and 95% confidence intervals for the association between the consumption of different types of fish (per 100 g/week) and incident type 2 diabetes (primary outcome) in men in the InterConnect project.

Cohort (Country)	Fatty Fish	Lean Fish	Seafood	Fried Fish	Salted, Dried Smoked, Fish	Saltwater Fish	Freshwater Fish
**Americas**							
ARIC (US)	1.04 (0.94, 1.16)	1.08 (0.99, 1.18)	1.00 (0.84, 1.17)				
ELSA-Brasil (Brazil)			0.94 (0.74, 1.19)	1.04 (0.92, 1.17)			
CARDIA (US)							
MESA (US)	0.86 (0.66, 1.12)	0.97 (0.80, 1.18)	0.98 (0.90, 1.08)	1.05 (0.85, 1.28)			
PRHHP (Puerto Rico)					1.07 (0.98, 1.16)		
WHI (US)							
**Eastern Mediterranean**							
Golestan (Iran)	1.01 (0.82, 1.25)	1.03 (0.91, 1.16)		1.08 (0.95, 1.22)	2.54 (0.04, 1.66)	1.02 (0.91, 1.14)	1.75 (0.84, 1.38)
**Europe**							
COSM (Sweden)	1.01 (0.98, 1.04)	1.01 (0.99, 1.04)	1.08 (1.01, 1.16)	1.04 (1.00, 1.08)			
EPIC-InterAct Denmark	0.99 (0.91, 1.07)	1.01 (0.91, 1.08)	1.02 (0.84, 1.24)	0.99 (0.90, 1.08)			
EPIC-InterAct France							
EPIC-InterAct Germany				0.93 (0.83, 1.05)			
EPIC-InterAct Italy							
EPIC-InterAct the Netherlands	0.72 (0.41, 1.28)	0.21 (0.08, 0.56)	0.22 (0.05, 1.11)	0.47 (0.29, 0.75)			
EPIC-InterAct Spain	1.00 (0.97, 1.04)	1.01 (0.99, 1.04)	1.03 (0.94, 1.12)	1.01 (0.79, 1.30)			
EPIC-InterAct Sweden	0.98 (0.93, 1.03)	1.10 (1.03, 1.17)	1.33 (1.16, 1.53)	0.95 (0.88, 1.03)			
EPIC-InterAct UK	0.93 (0.81, 1.06)	0.91 (0.80, 1.03)	1.26 (0.84, 1.89)	1.07 (0.81, 1.43)			
FMC (Finland)	0.95 (0.87, 1.05)	1.03 (0.97, 1.09)		1.02 (0.93, 1.11)	0.94 (0.84, 1.04)	0.95 (0.84, 1.06)	1.01 (0.96, 1.07)
Hoorn (the Netherlands)							
NOWAC (Norway)							
SUN (Spain)	0.97 (0.88, 1.07)	0.90 (0.93, 1.06)	0.97 (0.82, 1.15)		1.28 (0.50, 3.29)		
Whitehall II (UK)	0.99 (0.89, 1.11)	1.00 (0.89, 1.12)	0.98 (0.78, 1.22)	1.10 (0.94, 1.44)			
Zutphen Elderly (the Netherlands)							
**Western Pacific**							
AusDiab (Australia)				1.08 (0.90, 1.3)			
CKB (China)							
JPHC (Japan)	0.96 (0.92, 1.01)	0.91 (0.83, 1.00)	0.97 (0.87, 1.07)		1.01 (0.97, 1.06)		0.96 (0.92, 1.00)
NHAPC (China)			0.86 (0.65, 1.12)		1.05 (0.92, 1.20)	0.95 (0.76, 1.17)	0.94 (0.77, 1.14)
SMHS (China)			0.97 (0.93, 1.01)			1.00 (0.98, 1.02)	0.99 (0.97, 1.02)
**Overall IRR**	0.99 (0.98, 1.01)	1.01 (0.99, 1.04)	1.02 (0.97, 1.08)	1.01 (0.97, 1.06)	1.02 (0.98, 1.06)	1.00 (0.98, 1.01)	0.99 (0.97, 1.01)
**Heterogeneity**	*I*^2^ = 0%	*I*^2^ = 55%	*I*^2^ = 56%	*I*^2^ = 41%	*I*^2^ = 0%	*I*^2^ = 0%	*I*^2^ = 0%

*p* < 0.05. ARIC—Atherosclerosis Risk in Communities; ELSA-Brasil—Brazilian Longitudinal Study of Adult Health; CARDIA—Coronary Artery Risk Development in Young Adults Study; MESA—Multi-Ethnic Study of Atherosclerosis; PRHHP—Puerto Rico Heart Health Program; WHI—Women Health Initiative; FMC—Finnish Mobile Clinic Health Examination Survey; NOWAC—Norwegian Women and Cancer; COSM—Cohort of Swedish Men; SMC—Swedish Mammography Cohort; SUN—Seguimiento Universidad de Navarra (University of Navarra Follow-up); AusDiab—Australian Diabetes, Obesity and Lifestyle Study; CKB—China Kadoorie Biobank; JPHC—Japan Public Health Center-based; NHAPC—Nutrition and Health of Aging Population of China Study; SMHS—Shanghai Men Health Survey; SWHS—Shanghai Women Health Survey. Adjusted for age, sex (if applicable), education, smoking, physical activity, alcohol intake, BMI, comorbidities at baseline, energy intake, intake of fibre, red and processed meat, fruit, vegetables, and sugary drinks.

**Table 4 nutrients-13-01223-t004:** Adjusted incidence rate ratios and 95% confidence intervals for the association between the consumption of different types of fish (per 100 g/week) and incident type 2 diabetes (primary outcome) in women in the InterConnect project.

Cohort (Country)	Fatty Fish	Lean Fish	Seafood	Fried Fish	Salted, Dried Smoked, Fish	Saltwater Fish	Freshwater Fish
**Americas**							
ARIC (US)	1.01 (0.91, 1.12)	1.04 (0.98, 1.10)	0.95 (0.77, 1.17)				
ELSA-Brasil (Brazil)			1.09 (0.89, 1.34)	0.99 (0.84, 1.16)			
CARDIA (US)			1.01 (0.92, 1.10)				
MESA (US)	0.98 (0.77, 1.24)	1.06 (0.83, 1.35)	1,07 (0.81, 1.41)	0.83 (0.57, 1.22)			
PRHHP (Puerto Rico)							
WHI (US)	1.03 (1.00, 1.06)	1.00 (0.98, 1.03)	1.03 (0.99, 1.07)	1.14 (1.11, 1.17)			
**Eastern Mediterranean**							
Golestan (Iran)	0.88 (0.52, 1.47)	1.02 (0.89, 1.17)		1.08 (0.94, 1.25)	1.29 (0.02, 93.6)	0.97 (0.83, 1.13)	1.17 (0.91, 1.52)
**Europe**							
EPIC-InterAct Denmark	0.93 (0.84, 1.02)	1.10 (1.01, 1.19)	1.50 (1.15, 1.96)	0.95 (0.84, 1.08)			
EPIC-InterAct France	1.01 (0.84, 1.22)	1.00 (0.87, 1.14)	0.46 (0.36, 0.58)				
EPIC-InterAct Germany				1.09 (0.91, 1.30)			
EPIC-InterAct Italy			1.14 (0.98, 1.32)	1.46 (0.95, 2.24)			
EPIC-InterAct the Netherlands	1.32 (0.98, 1.78)	0.62 (0.39, 1.00)	1.78 (0.66, 4.81)	0.79 (0.62, 1.00)			
EPIC-InterAct Spain	1.13 (1.08, 1.19)	1.06 (1.03, 1.09)	1.18 (0.99, 1.26)	0.79 (0.53, 1.18)			
EPIC-InterAct Sweden	0.98 (0.92, 1.04)	0.98 (0.90, 1.06)	1.28 (1.09, 1.50)	1.00 (0.88, 1.13)			
EPIC-InterAct UK	1.07 (0.94, 1.22)	0.99 (0.89, 1.10)	1.45 (0.90, 2.32)	1.05 (0.77, 1.44)			
FMC (Finland)	1.06 (0.94, 1.20)	1.09 (1.00, 1.19)		1.09 (0.98, 1.21)	1.09 (0.95, 1.26)	0.99 (0.84, 1.15)	1.11 (1.03, 1.19)
Hoorn (the Netherlands)							
NOWAC (Norway)	1.04 (0.98, 1.12)	1.00 (0.96, 1.05)	1.09 (0.78, 1.52)				
SMC (Sweden)	1.01 (0.95, 1.07)	1.01 (0.98, 1.04)	1.06 (0.96, 1.18)	1.03 (0.98, 1.08)			
SUN (Spain)	1.12 (1.00, 1.24)	0.94 (0.81, 1.10)	1.06 (0.79, 1.43)		0.41 (0.03, 4.76)		
Whitehall II (UK)	1.03 (0.88, 1.19)	1.03 (0.87, 1.22)	0.97 (0.74, 1.27)	1.06 (0.67, 1.67)			
Zutphen Elderly (the Netherlands)							
**Western Pacific**							
AusDiab (Australia)				1.11 (0.97, 1.27)			
CKB (China)							
JPHC (Japan)	1.04 (0.99, 1.10)	1.04 (0.93, 1.16)	1.05 (0.94, 1.16)		1.02 (0.96, 1.09)		1.04 (0.99, 1.08)
NHAPC (China)			1.00 (0.81, 1.24)		0.92 (0.57, 1.49)	0.97 (0.82, 1.15)	0.99 (0.86, 1.14)
SWHS (China)			1.02 (0.98, 1.05)			1.00 (0.98, 1.02)	1.01 (0.99, 1.03)
**Overall IRR**	1.04 (1.01, 1.07)	1.02 (1.00, 1.04)	1.04 (0.98, 1.11)	1.04 (0.98, 1.10)	1.03 (0.97, 1.10)	1.00 (0.98, 1.02)	1.04 (1.00, 1.08)
**Heterogeneity**	*I*^2^ = 46%	*I*^2^ = 33%	*I*^2^ = 74%	*I*^2^ = 64%	*I*^2^ = 0%	*I*^2^ = 0%	*I*^2^ = 47%

*p* < 0.05. ARIC—Atherosclerosis Risk in Communities; ELSA-Brasil—Brazilian Longitudinal Study of Adult Health; CARDIA—Coronary Artery Risk Development in Young Adults Study; MESA—Multi-Ethnic Study of Atherosclerosis; PRHHP—Puerto Rico Heart Health Program; WHI—Women Health Initiative; FMC—Finnish Mobile Clinic Health Examination Survey; NOWAC—Norwegian Women and Cancer; COSM—Cohort of Swedish Men; SMC—Swedish Mammography Cohort; SUN—Seguimiento Universidad de Navarra (University of Navarra Follow-up); AusDiab—Australian Diabetes, Obesity and Lifestyle Study; CKB—China Kadoorie Biobank; JPHC—Japan Public Health Center-based; NHARC—Nutrition and Health of Aging Population of China Study; SMHS—Shanghai Men Health Survey; SWHS—Shanghai Women Health Survey. Adjusted for age, sex (if applicable), and education.

## Data Availability

The project was undertaken using a federated approach in which analyses are performed centrally while data remain within the governance structure of the original studies. Data are held peripherally across multiple research institutions rather than being transferred and stored at a central location. Researchers seeking the analysis dataset for this work should submit requests to the individual studies detailed in the supplementary information. Studies from the USA were downloaded from the BioLINCC repository on September 2016 (https://biolincc.nhlbi.nih.gov/home/, accessed on 16 March 2021).

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
