# Peer review of "Heterogeneity of Associations between Total and Types of Fish Intake and the Incidence of Type 2 Diabetes: Federated Meta-Analysis of 28 Prospective Studies Including 956,122 Participants"

_nutrients, 2021, doi:10.3390/nu13041223_

Round 1
Reviewer 1 Report
In the manuscript submitted by Pastorino et al. “Heterogeneity of associations between total and types of fish intake and the incidence of type 2 diabetes: federated meta-analysis of 28 prospective studies including 956,122 participants”, authors examined the association between total and types of fish consumption and type 2 diabetes, using individual participant data from 28 prospective cohort studies conducted in different countries. After having stratified all analyses by sex, they reported that in men all the associations were null, while in women they observed a modest positive association between total fish intake and type 2 diabetes, but with heterogeneity across different studies.
Major points
I have read the manuscript submitted by Pastorino et al. with interest and I believe that the purpose of expanding the knowledge on understanding the link between fish intake and incidence of type 2 diabetes is important and captivating. However, I have several major concerns about how the study has been conducted, the conclusion obtained and hence the take-home message given by the authors.
Authors analyzed a huge amount of data; however, they are extremely heterogenous under many points of view: time frames in which they were collected, participants’ demographics characteristics, kind of fish eaten, way of cooking/preserving it, dietary assessment methods, etc. Moreover, the majority of studies included in the analysis used food frequency questionnaires, which are not precise in quantifying food intake, rely on long-term memory and thus are often subject to measurement error. Not surprisingly, the results obtained were quite weak and inconclusive. Nevertheless, they could drive a message which can do far more harm than good: meaning that, in women, fish consumption may contribute to increase the risk of type 2 diabetes.
In my opinion, in order to significantly and valuably detect any connection between fish consumption and incidence of type 2 diabetes (or any kind of chronic disease), it is compulsory to examine, first of all a relevant amount of data, but, at the same time, those data must be as homogeneous as possible, in order to be able to draw meaningful conclusions. For instance: (i) do not mix different kind of cooking or preserving methods, (ii) differently consider farmed from wild fish (it wasn’t indicated in this study), (iii) do not mix data obtained from very different time frames or cultures, (iv) try to use data from food diaries, (v) use a design that allows for a clear examination of the temporal relationship between fish intake and type 2 diabetes development.
Moreover:
- Line 257: “Among types of fish there was a positive association between fatty fish consumption and type 2 diabetes incidence in women in the Americas (IRR 1.03, 95%CI: 1.001, 1.064; l2=0)”; actually, those data refer only to a specific cohort (WHI) grouped within the Americas category (Table 4). Thus, this sentence is misleading.
- Authors speculated that environmental contaminants might have contributed to the geographical and sex discrepancy in risk. However, they neither distinguish between farmed and wild fish consumption, nor consider the fact that a huge difference in the level of environmental contaminants does exist among the recruitment time frame of studies included in the analysis (for instance: 1967-1972, 1985-1986, 1991-1998, 2008-2010, 2000-ongoing, etc.). Moreover, 20 out of 29 studies were performed from 1967 to 1999.
Minor points
Line 91: type 2 diabetes (type 2 diabetes).
In conclusion, I do believe that this work has many methodological and conceptual flaws, which could wrongly pass the message that, in some cases, fish consumption may contribute to increase the risk of type 2 diabetes.
Reviewer 2 Report
The present manuscript describes a weak correlation between total fish intake and type 2 diabetes among women deriving data from a meta-analysis of several studies, which shows heterogeneities due to different eating habits matching with geographical location and types of fish intake. Pastorino et al meta-analysis was optimized and performed avoiding publication bias. This reviewer found of particular interest that findings revealed that fish consumption and type 2 diabetes associate only in women and not in men.
Following points should be addressed:
- Could the authors better comment the finding about sex association? Which is the hypothesis at the base of these results?
- Could the authors speculate about the results suggesting that other dietary inhabits according geographical area play a role in the association with type 2 diabetes?
- Table 4: please check size and format of the table.
Round 2
Reviewer 1 Report
I have read the authors reply to my first review report with interest and I have appreciated both the explanations given and the modifications/additional elucidations that they have made within the text.
I agree with authors, that the positive association they found between fish intake and T2D, in North American women, is consistent with prior literature. Moreover, I’ve found appropriated the fact that they specify in the Discussion (line 439-444) that this association might be due to other foods consumed in this particular geographic area.
Anyway, I still have some concern about their speculation on the association between fish consumption and T2D incidence in North American women.
Considering that:
- The typical Western diets (presumably the ones followed by the majority of North American women) may increase the risk to develop type 2 diabetes.
- The association they found between total fish intake and T2D are modest and so their results are not fully clear and require further investigation (as indeed stated by author themselves).
I think it is crucial, while waiting for further analysis/data/studies on this subject, to stress the message that the overall diet, rather than a single food component, could strongly affect the risk of developing diabetes. That’s why it is my opinion that stating in the conclusion part: “in some Western settings fish intake is unlikely to contribute to the prevention of type 2 diabetes, and may potentially marginally increase its future risk” could convey a misleading message, which, instead of recommending to go for a healthier diet (which includes regular fish consumption), could suggest to avoid this food component which, if cooked and consumed properly, particularly in a contest of a healthy diet, is anything but harmful for our health.
For this reason, I strongly recommend that, in the conclusion section of the paper, the authors clarify this point and the importance of an overall healthy diet (which do not exclude fish consumption) on the risk of T2D development.
